Non-invasive monitoring of glucocorticoid metabolite concentrations in urine and faeces of the Sungazer (Smaug giganteus)

Scheun Juan J.scheun@sanbi.org.za 1 2
Greeff Dominique 1
Ganswindt Andre 1 2
1 National Zoological Garden, South African National Biodiversity Institute , Pretoria , South Africa
2 Mammal Research Institute, Department of Zoology and Entomology, University of Pretoria , Pretoria , South Africa
Ringo John
Electronic publication date: 2018 Dec 21
Publication date: 2018
Volume: 6
Electronic Location ID: e6132
Received 2018 Sep 18; Accepted 2018 Nov 19
Copyright: ©2018 Scheun et al.
Copyright year: 2018
Copyright holder: Scheun et al.
License: This is an open access article distributed under the terms of the Creative Commons Attribution License, which permits unrestricted use, distribution, reproduction and adaptation in any medium and for any purpose provided that it is properly attributed. For attribution, the original author(s), title, publication source (PeerJ) and either DOI or URL of the article must be cited.
License URL: https://creativecommons.org/licenses/by/4.0/

Keywords: Validate, Endocrinology, Non-invasive hormone monitoring, Stress, Sungazer, Enzyme immunoassay, Faeces, Urine

Funding: The authors received no funding for this work.

==============================
Developing non-invasive techniques for monitoring physiological stress responses has been conducted in a number of mammal and bird species, revolutionizing field-based endocrinology and conservation practices. However, studies validating and monitoring glucocorticoid concentrations in reptiles are still limited. The aim of the study was to validate a method for monitoring glucocorticoid metabolite concentrations in urine (uGCM) and faeces (fGCM) of the cordylid lizard, the Sungazer (Smaug giganteus). An adrenocorticotropic hormone (ACTH) challenge was conducted on one male and two females with both urine and faecal material being collected during baseline and post-injection periods. Steroid extracts were analysed with four enzyme immunoassays (EIAs)namely: 11-oxoaetiocholanolone, 5α-pregnane-3β-11β-21-triol-20-one, tetrahydrocorticosterone, and corticosterone. A considerable response in fGCM and uGCM concentrations following ACTH administration was observed in all subjects, with the 5α-pregnane-3β-11β-21-triol-20-one and tetrahydrocorticosterone EIAs appearing to be the most suited for monitoring alterations in glucocorticoid metabolite concentrations in S. giganteus using faeces or urine as hormone matrix. Both EIAs showed a significantly higher concentration of glucocorticoid metabolites in faeces compared to urine for both sexes. Collectively, the findings of this study confirmed that both urine and faeces can be used to non-invasively assess adrenocortical function in S. giganteus.

Introduction

Historically, reptiles have been seen as a vertebrate group with limited importance to the natural environment, with the disappearance of the taxa unlikely to make any noteworthy difference (Zim & Smith, 1953). Thankfully, this sentiment has disappeared as scientists realize the importance of reptiles as an integral part of the ecosystem and thus important indicators of environmental quality (Gibbons & Stangel, 1999). Despite research showing that reptile numbers decline on a similar scale in terms of taxonomic breadth, geographic scope and severity as amphibians (Gibbons et al., 2000), reptiles remain one of the least studied vertebrate groups, being considered of less general interest compared to other fauna (Bonnet, Shine & Lourdais, 2002). The cryptic colorations and nature of reptiles (Zug, Vitt & Caldwell, 2001), as well as the general low population numbers inherent to the taxa (Todd, Willson & Gibbons, 2010), often results in a limited number of individuals available to monitor during a study. A number of factors have been suggested to contribute to the currently recognized decline in reptiles globally (see Todd, Willson & Gibbons, 2010 for a review). However, the direct and indirect effects of factors such as global climate change, disease, and habitat pollution are sometimes difficult to quantify and relate to individual and population health and survivability (Gibbons et al., 2000). In this regard, monitoring physiological stress patterns in reptiles may provide an important insight into the susceptibility of reptiles to population declines when faced with various environmental threats.

Stress is commonly referred to as the stimulus that may threaten, or appear to threaten, the general homeostasis of an individual (Wielebnowski, 2003; Hulsman et al., 2011). The perception of a stressor leads to the activation of the hypothalamic-pituitary-adrenal (HPA) axis and, consequently, to an increase in glucocorticoid (GC) secretion (Sapolsky, Romero & Munck, 2000; Hulsman et al., 2011). An acute increase in GC concentrations can be adaptive in nature, increasing energy availability and altering behavior, while indirectly regulating cardiovascular and metabolic parameters (Romero, 2002; Sapolsky, 2002; Reeder & Kramer, 2005; Walker, 2007). However, prolonged elevation of GC concentrations can lead to a number of deleterious effects, such as the suppression of the immune and reproductive systems, muscle atrophy, growth suppression, and a shortened life span (Möstl & Palme, 2002; Sapolsky, 2002; Charmandari, Tsigos & Chrousos, 2005; Cohen, Janicki-Deverts & Miller, 2007). Thus, monitoring GC concentration in endangered and threatened species can be an important tool for assessing physiological stress in individuals exposed to natural and anthropogenic stressors. Non-invasive hormone monitoring techniques, through the collection of urine or faeces, hold numerous advantages over the traditional use of blood collection. Firstly, there is no need to capture or restrain study animals for sample collection, thus removing any potential stress-related feedback, and thereby also increases safety for both animal subjects and researchers (Romero & Reed, 2005). Further, as a result of the general ease of collection, longitudinal sampling and hormone monitoring of specific individuals are possible (Heistermann, 2010). Finally, hormone metabolite concentrations determined from matrices like faeces, urine, or hair are usually less affected by episodic fluctuations of hormone secretion, as circulating hormone concentrations are accumulating in these matrices over a certain period of time (Vining et al., 1983; Creel, MarushaCreel & Monfort, 1996; Russell et al., 2012). However, prior to the first use of the chosen assays and specific matrices for monitoring physiological stress in a species, it is important that the approach is carefully validated to ensure a reliable quantification of respective GCs (Touma & Palme, 2005). A preferred method of validation is the physiological activation of the HPA axis through the injection of adrenocorticotropic hormone (“ACTH challenge”, (Touma & Palme, 2005), which results in a distinct increase in GC production from the adrenal gland. Collected pre- and post-injection samples are subsequently analyzed to determine which of the tested enzyme immunoassays (EIA) reflects the induced increase in GC concentrations best. Historically, GC patterns in reptiles have been monitored via serum analysis, for example in the red-eared slider turtle (Trachemys scripta elegans, Cash, Holberton & Knight, 1997), the Galapagos marine iguana (Amblyrhynchus cristatus, Romero & Wikelski, 2001), or the tuatara (Sphenodon punctatus, Tyrrell & Cree, 1998). However, some more recent studies attempting to understand the physiological response inherent in reptiles to environmental stressors have already opted for non-invasive hormone monitoring in reptiles, e.g., in Nile crocodiles (Crocodylus niloticus, Ganswindt et al., 2014), the three-toed box turtle (Terrapene carolina triunguis, Rittenhouse et al., 2005), the green anole (Anolis carolinensis, Borgmans et al., 2018) or the green iguana (Iguana iguana, Kalliokoski et al., 2012).

The Sungazer (Smaug giganteus, formerly Cordylus giganteus; Fig. S1) is a cordylid lizard endemic to the grassland of the Free State and Mpumalanga provinces of South Africa (De Waal, 1978; Jacobsen, 1989). It is unique among the cordylid lizards as an obligate burrower rather than rupicolous (Tonini et al., 2016; Parusnath et al., 2017). The species is currently facing large scale habitat degradation and population declines as a result of anthropogenic activities such as agricultural repurposing of its natural habitat, road construction, electricity infrastructure, mining developments, as well as the pet and traditional medicine trade (Van Wyk, 1992; McIntyre & Whiting, 2012; Mouton, 2014). Consequentially, S. giganteus is now listed as vulnerable by the International Union for the Conservation of Nature (IUCN, Alexander et al., 2018).

The aim of the study was to examine the suitability of four enzyme immunoassays (EIA) namely, 11-oxoaetiocholanolone, 5α-pregnane-3β-11β-21-triol-20-one, tetrahydrocorticosterone, and corticosterone, for monitoring adrenocortical function in S. giganteus by determining the stress-related physiological response in faeces and urine following an ACTH challenge test.

Materials & Methods

Study site and animals

The study was conducted at the SANBI National Zoological Garden (NZG), Pretoria, South Africa (25.73913°N, 28.18918°E) from the 24th of November 2017 to the 5th of December 2017. The study animals, consisting of one male (M1:291 g) and two females (F1: 295 g) and F2: 344 g), were housed in individual enclosures within the Reptile and Amphibian Section of the NZG. Individuals were separated by a 1.5 m high wall, which resulted in study animals not being in visual contact with one another. Each enclosure (2 m × 1.5 m) was covered in coarse river sand and included an artificial burrow constructed from fiberglass, UV-light and a water bowl with water available ad libitum. The light regime (13 L: 11 D) and humidity (range: 44–50%) were kept constant throughout the study period. A combination of meal worms and fresh, green vegetables were provided daily to all individuals. Prior to the start of the study, all individuals were given a two-week acclimatization period to the new enclosure and presence of researchers. The limited number of individuals used during the study reflects the availability of study animals in a suitable setting, as well as the difficulty in receiving approval to conduct research on vulnerable and endangered species.

Sample collection and ACTH challenge

In reptiles, urine and faeces can be excreted in unison, though not mixed (Fig. S2; Singer, 2003); urine is a white, solid substance, compared to the dark, solid faecal component, which allows for the separation of the two matrices with limited levels of cross-contamination (Kummrow et al., 2011). During the entire monitoring period, collected urine and faeces were separated during collection, and the two parts placed into separate 1.5 ml microcentrifuge tubes, sealed, and immediately stored at −20 °C until further processing. Following a two-week acclimatisation period, enclosures were checked for urine and faecal samples during the active period of S. giganteus (6 am–6 pm), for seven days. Cages were checked hourly to limit the effect of bacterial and environmental degradation of urine and faecal samples. In the morning hours of the eighth day all three individuals were injected intramuscularly with 0.45 µg synthetic ACTH g−1 bodyweight (SynACTH®, Novartis, South Africa Pty Ltd) in a 100 µl saline transport. This ACTH dose was chosen as it has been used successfully by a number of studies conducted on amphibian species such as the Fijian ground frog (Platymantis vitiana, Narayan et al., 2010), tree frog (Hypsiboas faber, Barsotti et al., 2017) and the American bullfrog (Rana catesbeiana, Hammond et al., 2018) to evoke a stress response. Subsequently, the individuals were released back into their individual enclosures, with faecal and urine collection continuing until day 15 of the study. The study was performed with the approval of the National Zoological Garden’s Animal Use and Care Committee (Reference: P16/22).

Steroid extraction in urine and faecal samples

Urine and faecal samples were lyophilized, pulverized and sifted through a mesh strainer to remove any undigested material, resulting in a fine faecal and urine powder (Heistermann, Tari & Hodges, 1993). Subsequently, 0.050–0.055 g of the respective urine and faecal powder was extracted with 1.5 ml 80% ethanol in water. After vortexing for 15 min, the suspensions were centrifuged for 10 min at 1,600 g and the resulting supernatants transferred into new microcentrifuge tubes and stored at −20 °C until analysis.

Enzyme immunoassay analyses

Depending on the original matrix, steroid extracts were measured for immunoreactive faecal glucocorticoid metabolite (fGCM) or urinary glucocorticoid metabolite (uGCM) concentrations using four different EIAs: (i) an 11-oxoaetiocholanalone (detecting fGCMs with a 5β-3α-ol-11-one structure), (ii) a 5α-pregnane-3β-11β-21-triol-20-one (measuring 3β-11β–diol-CM), (iii) a tetrahydrocorticosterone, and (iv) corticosterone EIA. Details about assay characteristics, including full descriptions of the assay components including cross-reactivities, can be found in Möstl et al. (2002) for the 11-oxoaetiocholanalone EIA, Touma et al. (2003) for the 5α-pregnane-3β-11β-21-triol-20-one, Palme & Möstl (1997) for the corticosterone EIA and in Quillfeldt & Möstl (2003) for the tetrahydrocorticosterone EIA. Assay sensitivities, which indicates the minimum amount of respective immunoreactive hormone that can be detected at 90% binding, as well as the intra- and inter-assay coefficients of variation of high and low quality controls for each EIA is shown in Table 1. Serial dilutions of extracted faecal and urine samples gave displacement curves that were parallel to the respective standard curves in the two assays of choice (5α-pregnane-3β-11β-21-triol-20-one and tetrahydrocorticosterone EIAs), with a relative variation in slope of <4%. All EIAs were performed at the Endocrine Research Laboratory, University of Pretoria, South Africa, as described previously (Ganswindt et al., 2002).

Table 1 The enzyme immunoassay specific parameters used during this study.

The sensitivity as well as the intra- and inter-assay coefficient of variation (CV) of the four enzyme immunoassays used during the study.

Enzyme immunoassay	Sensitivity (ng/g dry weight)	Intra-assay CV	Inter-assay CV	
11-oxoaetiocholanalone	2.4	4.24% & 5.31%	1.69% & 7.06%	
5α-pregnane-3β-11β-21-triol-20-one	0.6	6.62% & 6.70%	7.33% & 9.79%	
Tetrahydrocorticosterone	9.0	6.33% & 6.64%	11.94% & 14.20%	
Corticosterone	1.8	4.15% & 5.41%	13.94% & 14.58%	

Data analysis

A total of six faecal and urine samples were analyzed for each individual. Individual median fGCM and uGCM concentrations from pre-injection samples were calculated, reflecting individual baseline concentrations. To determine the effect of the ACTH injection on the HPA axis, the fGCM and uGCM concentrations from post-injection samples were converted to percentage response, by calculating the quotient of individual baseline and related fGCM/uGCM samples. In this regard, a 100% (1-fold) response represents the baseline value.

Furthermore, the mean absolute deviation (MAD) was calculated for the baseline sample set (pre-injection). The MAD of the particular dataset shows the average distance between each baseline period data point and the calculated mean thereof, which represents the variability of the baseline samples collected. Thus, the lower a MAD value is for a specific EIA, the more stable the assay. Here, the individual baseline uGCM/fGCM concentration was subtracted from all pre-injection fGCM/uGCM values for each EIA-specific data set. The differences were noted as absolute values and the mean of the absolute values calculated, representing the MAD value for each EIA. The MAD values were converted to a percentage deviation value (MAD/Baseline Value*100) to allow for the comparison between EIAs. To determine the effect of the ACTH injection, the absolute change in fGCM and uGCM concentration was determined by calculating the quotient of baseline and post-injection peak fGCM and uGCM samples. MAD values below 15% were regarded as preferable.

The most appropriate EIA for measuring fGCM and uGCM concentrations in the species was chosen by comparing (1) the highest post-injection signal and (2) lowest MAD values observed. Values are given as mean ± standard deviation (SD) where applicable. Analytical statistics and graphical designs were performed using R software (R 3.2.1; R Development Core Team, 2013).

Results

Defecation rate and MAD results

The average defecation rate (time between defecation events) showed considerable variation between individuals and matrices (Table 2).

Table 2 Urinary and faecal excretion rate, along with the time to peak urinary and faecal glucocorticoid metabolite peaks.

The average faecal and urine excretion rate for female and male individuals of the study. Time to peak fGCM and uGCM response, as well as the respective sample numbers, are shown for each study animal. Values are given as mean ± standard deviation.

ID	Faecal excretion rate Hours	Urine excretion rate hours	Time to peak fGCM sample post-injection hours+ (sample number)	Time to peak uGCM sample post-injection hours+ (sample number)	
Female 1	56.6 ± 39.9	38.8 ± 30.8	24 (1)	27 (1)	
Female 2	57.8 ± 20.0	39.8 ± 21.3	24 (1)	27 (1)	
Male	37.8 ± 36.1	43.2 ± 30.8	105 (1)	97 (3)	

The percentage MAD values were considerably lower in all EIAs when analyzing faecal (range: 3.17–15.67%) compared to urine (range: 13.31–56.52%) samples. For faeces, although the corticosterone EIA showed the lowest average percentage MAD value (mean ± SD = 8.37 ± 5.54%), the remaining three EIAs all showed comparable low average MAD levels (range: 9.95–11.02%). In contrast, all four EIAs showed high average percentage MAD levels in urine, with the 11-oxoaetiocholanalone EIA having the lowest average MAD value (mean ± SD = 17.17 ± 5.77%).

Faecal glucocorticoid metabolite analyses

All four EIAs showed a considerable response (178.46–744.84%) in fGCM concentration following the ACTH injection in all three study animals (Table 3). For F1, the 5α-pregnane-3β-11β-21-triol-20-one and tetrahydrocorticosterone EIAs showed the highest response, exceeding 450%, at 24 h post ACTH administration (Tables 2 and 3, Fig. 1A). Following this, fGCM concentrations did not return to baseline levels during the monitoring period. For F2, the tetrahydrocorticosterone and corticosterone EIA performed best, with responses exceeding 300%, at 24 h post ACTH injection (Tables 2 and 3, Fig. 1B). The fGCM concentrations returned to baseline levels 80 h post ACTH administration for F2. For M1, the 5α-pregnane-3β-11β-21-triol-20-one EIA showed the highest response, exceeding 700%, at 105 h post ACTH administration (Tables 2 and 3, Fig. 1C). Similar to F2, the respective fGCM concentrations returned to baseline levels 120 h post ACTH administration (Fig. 1C).

Table 3 The urinary and faecal glucocorticoid metabolite response following ACTH administration in Smaug giganteus.

The peak percentage glucocorticoid response in both faeces and urine, across all four enzyme immunoassays tested, in the two female and one male individual following the adrenocorticotropic hormone challenge. Values are given as mean ± SD.

ID	Matrix	11-oxoaetiocholanolone (%)	5α-pregnane-3 β-11β-21-triol-20-one (%)	Tetrahydrocorticosterone (%)	Corticosterone (%)	
F1	Faeces	230.4	697.5	461.9	214.6	
F2	Faeces	178.5	219.5	436.6	317.6	
M1	Faeces	262.6	744.8	546.6	307	
Mean ± SD	223.8 ± 42.4	554 ± 290.6	481.7 ± 57.6	279.7 ± 56.7	
F1	Urine	149.5	341.5	651.8	246.9	
F2	Urine	159.7	210.3	554	239.6	
M1	Urine	391.3	347.4	347.9	195.3	
Mean ± SD	233.5 ± 136.8	299.7 ± 77.5	517.9 ± 155.2	227.3 ± 27.9	

Figure 1 The faecal glucocorticoid metabolite response in the study animals following ACTH administrations.

The percentage fGCM response displayed by each of the four tested enzyme immunoassays for F1 (A), F2 (B) and the male (C), following ACTH administration (time 0). Pre-injection baseline values were used as reference concentrations and set as 100%.

Urinary glucocorticoid metabolites analysis

Similar to the fGCM findings, all four EIAs showed a considerable response in uGCM concentrations (149.53%–651.82%) following the ACTH injection (Table 3). For the two females, the 5α-pregnane-3β–11β–21-triol-20-one and tetrahydrocorticosterone EIA showed the highest response, exceeding 340%, in the first collected faecal sample 27 h post ACTH administration (Tables 2 and 3, Fig. 2A & Fig. 2B). Respective uGCM concentrations returned to baseline levels at 51 h and 80 h post ACTH administration for F1 and F2, respectively. In contrast to the female profiles, the highest response to the ACTH administration in the male was found using the 11-oxoaetiocholanalone EIA (391.37% %), with both the 5α-pregnane-3β-11β-21-triol-20-one and tetrahydrocorticosterone EIAs also showing suitable responses (347.36%–347.87%; Fig. 2C). The peak response occurred 97 h post ACTH injection (Table 2), and thus male uGCM concentrations did not return to baseline levels during the monitoring period.

Figure 2 The urinary glucocorticoid metabolite response for all study animals following ACTH administration.

The percentage uGCM response displayed by each of the four tested enzyme immunoassays for F1 (A), F2 (B) and the male (C) 1, following ACTH administration (time 0). Pre-injection baseline values were used as reference concentrations and set as 100%.

Discussion

In the current study, the defecation rate of study animals were prolonged and varied substantially within and between individuals. Extended defecation rates have been observed in a number of reptile species such as the Italian wall lizard (Podarcis sicula, ∼50 h, (Vervust et al., 2010), veiled chameleon (Chamaeleo calyptratus, ∼96 h, (Kummrow et al., 2011), six striped runner (Cnedmidophurs sexlineatus, 23–26 h, (Hatch & Afik, 1999) and a variety of snake species (45–3,180 h, (Lillywhite, De Delva & Noonan, 2002). Additionally, these studies have also shown high levels of individual variability in terms of gut retention times; for example, Kummrow et al. (2011) observed an individual excretion rate in C. calyptratus ranging from 48–120 h, while Hatch & Afik (1999) found the excretion rate in C. sexlineatus to range from 20–72 h. Understanding species-specific differences and individual variability in faecal and urinary defecation rates are important for a number of reasons. Firstly, the infrequent and extended excretion rate of urinary and faecal material in reptiles may complicate data interpretations (Ganswindt et al., 2014). Furthermore, the movement of urine into the cloaca (urodeum) before moving into the intestines, where urinary and faecal material can be excreted in unison (Singer, 2003), can further complicate the distinction between matrix-specific retention time and steroid hormone metabolite excretion routes. As it is difficult to collect frequent faecal and urine samples consistently in S. giganteus and other reptile species, it may be advisable to monitor GC metabolite patterns over a longer time period to successfully determine the possible effect a defined stressor may have on an individual or species (Kummrow et al., 2011).

The MAD values for the four EIAs used in the fGCM analysis indicated low levels of variation from the predetermined baseline values. In contrast to this, the MAD values calculated for the four uGCM EIAs showed high levels of variation from calculated baseline levels. As such, GC metabolite excretion via faeces may be less prone to regular fluctuation than urine, although further research is required to confirm this.

Following ACTH injection, the peak fGCM response was observed in the first faecal sample collected from all study animals. Ganswindt et al. (2014) found peak fGCM concentration, following the ACTH injection, in the first collected faecal sample from C. niloticus. Similarly, Cikanek et al. (2014) observed peak fGCM concentrations in the first collected faecal sample in Harlequin frogs (Atelopus certus) following a biological stressor. The pooling of faecal material in the reptile gut, over an extended period of time, may explain why peak fGCM responses are observed in the first sample post-injection in reptiles and other infrequent defecators. However, the available literature on reptile fGCM monitoring is limited, with a number of studies failing to highlight when the peak fGCM levels were observed or choosing to pool samples into larger time periods (Rittenhouse et al., 2005). Although all four EIAs displayed considerable peak fGCM responses for both sexes, the tetrahydrocorticosterone and 5α-pregnane-3β-11β-21-triol-20-one EIA performed best in our study, based on (i) EIA stability as seen in the low MAD values and (ii) the magnitude of peak percentage fGCM response following the ACTH injection. As such, both EIAs seem to be suitable for monitoring alterations in fGCM concentration in S. giganteus faecal material.

The peak uGCM concentrations following ACTH administrations were observed in the first and third collected urine sample for the females and male respectively. To our knowledge this is the first study to quantify the uGCM response following the activation of the HPA axis through physiological or biological stressors. In reptiles, the movement of urine into the intestine, and the resulting pooling effect along with faeces over time, may explain why peak uGCM responses were observed within the first collected samples for females and third sample for males. Similar to the fGCM analysis, all four EIAs used during the study were able to monitor alterations in uGCM concentrations following the ACTH administration; the tetrahydrocorticosterone and 5α-pregnane-3β-11β-21-triol-20-one EIAs again showed the highest uGCM response in this regard. With all uGCM MAD values considerably higher than observed for the fGCM analysis, the peak uGCM response values were used to determine EIA suitability; in this regard, both the tetrahydrocorticosterone and 5α-pregnane-3β-11β-21-triol-20-one EIAs were deemed suitable for monitoring alterations in uGCM concentration in S. giganteus urine.

Conclusion

The ability to monitor physiological stress patterns in endangered reptile species, through non-invasive hormone monitoring techniques, offers conservationists an ideal tool which can be implemented within both free-ranging and captive setups with limited effort. With the increase in human-driven factors leading to substantial decreases in reptile populations, the need for such techniques are becoming more important. This study has successfully validated such a technique for monitoring the stress response in S. giganteus in both urine and faeces by using the 5α-pregnane-3β-11β-21-triol-20-one or tetrahydrocorticosterone EIA. Both assays showed low MAD values as well as a considerable response in fGCM and uGCM concentrations following ACTH injection. As such, both sample matrices can be used to monitor physiological stress in S. giganteus. Despite the results of this study, a number of uncertainties need to be addressed by researcher conducting further studies on the topic. Of greatest concern is the observed gut passage time and time to peak fGCM and uGCM concentrations between individuals. Although the time to peak fGCM (24 h) and uGCM (27 h) responses were similar in both females, the monitored male showed a prolonged gut passage time with peak fGCM and uGCM concentrations 81 h and 70 h later, respectively. However, if in fact differences in gut passage time or GC metabolite patterns between individuals or sexes of the species exist is yet to be determined by examining larger study populations. Currently, we recommend collecting only the faecal or urine component for GC metabolite monitoring in S. giganteus. Despite the limitations of this study the findings increased our understanding of stress hormone production, metabolism and excretion pattern in the species. We hope this will encourage and stimulate future research not only on this species, but reptiles in general, especially concerning the non-invasively examining the physiological stress response linked to a host of anthropogenic and natural factors.

Supplemental Information

Figure S1 Smaug giganteus (Photographer: Juan Scheun)

Click here for additional data file.

Figure S1 Smaug giganteus urofaecal sample

The white component represents urine, while the dark section represents faeces (Photographer: Juan Scheun)

Click here for additional data file.

We would like to thank the staff of the NZG reptile and veterinary sections for their assistance throughout the study. Additionally, we would like to thank Abongile Ndzungu at the Endocrine Research Laboratory for expert help in laboratory techniques.

Additional Information and Declarations

Competing Interests

Author Contributions

Animal Ethics

Data Availability

The authors declare there are no competing interests.

Juan Scheun conceived and designed the experiments, performed the experiments, analyzed the data, contributed reagents/materials/analysis tools, prepared figures and/or tables, authored or reviewed drafts of the paper, approved the final draft.

Dominique Greeff performed the experiments, analyzed the data, authored or reviewed drafts of the paper, approved the final draft.

Andre Ganswindt conceived and designed the experiments, analyzed the data, contributed reagents/materials/analysis tools, authored or reviewed drafts of the paper, approved the final draft.

The following information was supplied relating to ethical approvals (i.e., approving body and any reference numbers):

National Zoological Garden Animal Use and Care Committee approved the research.

The following information was supplied regarding data availability:

Scheun, Juan. (2018). Sungazer urinary and faecal glucocorticoid metabolite concentrations [Data set]. Zenodo. http://doi.org/10.5281/zenodo.1744809.

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
