# Peer review of "Non-invasive monitoring of glucocorticoid metabolite concentrations in urine and faeces of the Sungazer (Smaug giganteus)"

_PeerJ, doi:10.7717/peerj.6132_

## Round 0.1 · original submission · Minor Revisions

Both the reviewers and I agree that your paper will be publishable after minor revision. Reviewer #1 wants more changes, probably because they took more care in the details of your report. I look forward to receiving the revised manuscript, after you've made the changes the reviewers request.

Reviewer 1 ·

Basic reporting

The basic reporting of the manuscript by Scheun et al. is mostly clear and coherent. The structure of the manuscript is sound, and the contents are mostly easily absorbed. The raw data has been made available.

Experimental design

The experimental design is sound with respect to the aims of the study. The low number of subjects (3) mixing both sexes (2 + 1) undermines the study somewhat, but I know from personal experience how hard these studies can be to set up and justify for ethics committees. The research question is somewhat clear, but the precise hypothesis could be more explicit (what makes the ideal assay?). Moreover, the methods need a bit more detail for someone to be able to replicate them.

Validity of the findings

The findings are fairly clear, although some of the data analysis seems excessive (for lack of a better word). The use of arbitrary numberings for fecal/urinary samples, as opposed to providing timing is weird and leads to some odd analyses. Given that this is a study that will serve as a future guide for researchers studying stress in sungazers, I am also missing some clear recommendations in the conclusions.

Additional comments

Scheun and colleagues have made a thorough investigation/validation of how to non-invasively assess stress in sungazers. This experiment is key for future studies looking into factors affecting stress in the species. There is, as the authors state, much less work done on reptiles than there is for mammalian species, and with their somewhat cryptic and passive behavior, biomarker-based assessments are all the more important.

Reading the report you get the sense that the authors are somewhat impatient in getting these findings out the door; no doubt because they have other sungazer studies planned. Whereas the experiments seem solidly designed and well-conducted, the reporting could be better. The introduction needs work, the methods are missing some information, the purpose of some of the data treatment is unclear, some of the results are reported in odd opaque ways, and the paper would really benefit from some clear recommendations included in the conclusions.

Specific comments (ordered by section):

Abstract
Line 25: Based on one male and two females, no conclusions with respect to sex-specific effects can be drawn. Substitute “was observed for both sexes” with “was observed in all subjects.”

Introduction
Line 45: I realize that the frequency of studies in reptiles is lower than for most mammalian species but calling a 14 years old paper “recent” and an 18 year old one “new” is pushing it. Find newer references or rephrase the sentence.
Line 50: This sentence needs more context. We have trouble studying them because ... we cannot find them?
Line 64: To my knowledge, GCs do not affect cardiovascular activity (depending on what is meant by enhance, I suppose) and the references do not support this statement.
Line 70: “...monitoring GC concentrations in e.g. endangered species can therefore not only assist in defining the presence and effect of local stressors, but also provide an argument for appropriate conservation actions if necessary.” What does this sentence mean? Rephrase for a clearer message.
Line 82: What is meant by “respective” here?
Line 96: How does its burrowing preference “add to” its genetic distinctiveness?
Line 96: It is not number 19 in total; it is in place 19 among the threatened ones. (What an odd factoid...)
Line 98: Why “despite?” Are not endemic creatures more vulnerable to habitat destruction as a rule?
Line 100: I am assuming that pet and medicine trades are not indirectly affecting the animals, but that sungazers are sold as pets and used in traditional (?) medicine? This sentence needs work.
Line 106: What determines the suitability of an assay in this regard? What, exactly, did the authors set out to find? What are the hallmarks of an ideal assay in this context?

Materials & methods
Throughout: Units should be separated by number with a blank space (SI standard). For centrifuges, “g” is used as a unit and consequently the multiplication sign should be removed.
Line 117: Could the animals see one another?
Line 119: Substrate, both for the cage and with respect to the artificial burrow should be listed.
Line 120: What is listed in the parenthesis is not light intensity (which, if measured, should be listed in lux), but the light regimen.
Line120: The humidity data seem off. Average +/- 1 SD should for an ideal distribution encompass 68% of data. Yet, here it seems to encompass 100%. Either the range is listed incorrectly, or the SD is calculated in a truly weird way.
Line 134: Were samples collected hourly around the clock? The raw data sheet only lists sample collections in the time interval of 07:00 and 15:00. Do sungazers only poop during working hours?
Line 136: There is no mention of discarded samples in either results, nor in the raw data. This is important to list! Particularly with respect to the arbitrary ordering of samples.
Line 139: The listed ACTH dose is incorrect/makes no sense. Synacthen comes in ampoules of 250 µg, which means that the authors used about 500 ampoules per animal if the dose was 0.433 mg/g... 1-2 IU/kg is more reasonable, but it is highly inexact. What was the actual dose? Two of the listed references – Kindermann et al., Graham et al. – report the same bizarre dose (some weird math has been going on here) whereas Cartledge & Jones use a fixed dose (independent of body weight) of 25 µg (= 2.5 IU). Narayan et al. do not use ACTH in their study or even mention it!
Line 149: Sample treatment could be better explained. Moreover, the reference – Feiß et al. – does not explain the method, but in turn references Heistermann et al. (1993).
Line 166: What is meant by “sensitivity” in this context? LLOQ? The authors need to properly define this.
Line 166: Detection limits, Intra/inter-assay CVs, and parallellisms for the four assays are well suited for being reported in a table.
Line 184: Was 100% defined relative to (the absolute) zero or to the detection limit?

Results
Line 207: “The average defecation rate (number of defecations per time unit) for urine and faeces exceeded 37 hours per defecation event for the three study animals...” The authors start with a perfectly sensible way of expressing defecation – “number of defecations per time unit”, then promptly throw it out the window and report “37 hours per defecation event.” What does that even mean? Is that the average time between two defecations? If it is, does this account for “non-fresh” samples being thrown out? According to the data in Table 1 these variables are normally distributed. Defecation rate has been highly variable in the reptiles I have worked with – are the authors sure that data are well-described using means and standard deviations? (The raw data sheet also suggests sungazers are fairly irregular...)
Line 209 & 254: What can we learn from the subject-level tests? The authors have only three subjects and can thus not establish a good population (even if they had all been of the same sex). What would be the interpretation if there were significant differences? This feels like a hypothesis-less hypothesis test.
Line 213: What do we learn from the MAD values? Are we looking for a low or high value? If there is a circadian rhythm to uGCM or fGCM we would expect this to show as increased MAD. So is that a bad thing? The data are not presented in full either. It is as if the authors themselves do not know what to make of it.
Line 222: What are we looking for when comparing assays? Highest signal or lowest variability?
Line 230: “...both the 5α-pregnane-3β-11β-21-triol-20-one and tetrahydrocorticosterone showed the highest response...” No, one of them showed a higher signal than the other.
Line 251: What can we learn from these correlations? Which samples are correlated with one another? (And importantly: Why?!) The urine and fecal samples are not obtained at the same time. If urine sample 3 and fecal sample 3 are a matched x-y-pair, we are comparing samples that can be as much as three days apart in time. What is the point here?

Discussion
Line 259: “...the validity of two EIAs, for monitoring a physiological stress response in the faeces and urine of S. giganteus, have been confirmed...” Which ones? And based on what? This comment is getting ahead of itself.
Line 262: “The defecation rate of a species is a relevant estimate for the delay in hormone metabolite excretion in faeces...” Irrespective of intestinal passage time? Explain. Also, the reference – Palme et al. (1996) – specifically does not discuss defecation rate, but spends quite a lot of time discussing intestinal passage time.
Line 280: “...it may be advisable to interpret a likely ”cause-and-effect” relationships over a longer time period.” I do not understand what the authors are trying to say. Sentence should be elaborated on.
Line 284: “The MAD values for the four EIAs used in the fGCM analysis indicated low levels of variation from the predetermined baseline values and thus EIA stability.” What is a low MAD value? Is 5% low? 10%? What is a good MAD? And what is meant by “EIA stability.”

Conclusions
The conclusions should list clear recommendations to researchers looking to investigate stress in sungazers. What assay should be employed and why? Should they collect fecal samples, urine, or both; and why? How long after a stress-response should they expect to see an elevation in their chosen assay? What could possibly confound their analyses and interpretations of these analyses?
Figures 1 & 2: These figures are weird since the samples are along an arbitrary x-axis. For example: The fGCM data points at x-axis position “1” correspond to 24 hours after ACTH injection for subjects F1 and F2, but to 105 hours after injection for M1. Why are these three points grouped together? The x-axis should show the time and any discarded samples need to be acknowledged.
Tables 1 & 2: Are the values really so accurately obtained as to be usefully reported down to two decimal places?

Reviewer 2 ·

Basic reporting

No comment

Experimental design

Line 116: The date of study is future dated !!!
Line 129: Please provide a photo showing the study species and the excreta with distinction between urine and faeces shown.
Line 139: Did the authors consider sampling the lizards using control stressors such as manual restraint or saline only injection?

Validity of the findings

Line 139: It cannot be confirmed without control groups whether the responses observed were due to the ACTH or the needle or handling. Please clarify.

Additional comments

A very interesting and important study validating stress hormone evaluation in an endangered reptile species, the Sungazer. I recommend acceptance with minor corrections.

---

## Round 0.2 · accepted · Accept

I appreciate your detailed rebuttal, noting and explaining the revisions.
Nicely done report!

#